# High Resolution Air Quality Forecasting over Prague within the URBI PRAGENSI Project: Model Performance during the Winter Period and the Effect of Urban Parameterization on PM

**Jana Ďoubalová** [1,2,*] , **Peter Huszár** [2] , **Kryštof Eben** [3], **Nina Benešová** [1], **Michal Belda** [2] , **Ondřej Vlček** [1] , **Jan Karlický** [2,4], **Jan Geletič** [3] **and Tomáš Halenka** [2]

1   Czech Hydrometeorological Institute, Na Šabatce 17, 14306 Prague 4, Czech Republic; nina.benesova@chmi.cz (N.B.); ondrej.vlcek@chmi.cz (O.V.)
2   Department of Atmospheric Physics, Faculty of Mathematics and Physics, Charles University, V Holešovičkách 2, 180 00 Prague 8, Czech Republic; peter.huszar@mff.cuni.cz (P.H.); michal.belda@mff.cuni.cz (M.B.); jan.karlicky@mff.cuni.cz (J.K.); tomas.halenka@mff.cuni.cz (T.H.)
3   Institute of Computer Science of the Czech Academy of Sciences, Department of Complex Systems, Pod Vodárenskou věží 271/2, 182 07 Prague 8, Czech Republic; eben@cs.cas.cz (K.E.); geletic@cs.cas.cz (J.G.)
4   Institute of Meteorology and Climatology, Department of Water, Atmosphere and Environment, University of Natural Resources and Life Sciences, Vienna, Gregor-Mendel-Straße 33, 1180 Vienna, Austria
*   Correspondence: jana.doubalova@chmi.cz

**Abstract:** The overall impact of urban environments on the atmosphere is the result of many different nonlinear processes, and their reproduction requires complex modeling approaches. The parameterization of these processes in the models can have large impacts on the model outputs. In this study, the evaluation of a WRF/Comprehensive Air Quality Model with Extensions (CAMx) forecast modeling system set up for Prague, the Czech Republic, within the project URBI PRAGENSI is presented. To assess the impacts of urban parameterization in WRF, in this case with the BEP+BEM (Building Environment Parameterization linked to Building Energy Model) urban canopy scheme, on Particulate Matter (PM) simulations, a simulation was performed for a winter pollution episode and compared to a non-urbanized run with BULK treatment. The urbanized scheme led to an average increase in temperature at 2 m by 2 °C, a decrease in wind speed by 0.5 m s$^{-1}$, a decrease in relative humidity by 5%, and an increase in planetary boundary layer height by 100 m. Based on the evaluation against observations, the overall model error was reduced. These impacts were propagated to the modeled PM concentrations, reducing them on average by 15–30 μg m$^{-3}$ and 10–15 μg m$^{-3}$ for PM$_{10}$ and PM$_{2.5}$, respectively. In general, the urban parameterization led to a larger underestimation of the PM values, but yielded a better representation of the diurnal variations.

**Keywords:** air pollution; emissions; urban canopy; weather prediction; particulate matter; validation

## 1. Introduction

The increasing urban population worldwide requires elaborate scientific action to estimate the human exposure to different adverse effects that cities cause [1]. It is now well known that probably the most "far-reaching" impact that cities have on the environment is that on the atmosphere [2].

Urban areas impact the atmosphere by two main pathways: they have a strong impact on meteorological conditions, which results in the formation of, e.g., Urban Heat Islands (UHI), and they are, at the same time, intense concentrated emission sources, having great impacts on the local and regional (and even global) air quality [3,4] with (although minor) consequences even on the regional

radiative balance and, hence, temperatures [5]. Given the complexity and non-linear nature of chemical transformations of primary emissions forming secondary species, modeling approaches are essential to describe the city-scale air quality conditions and their footprints over a regional scale. Such modeling approaches have to describe both the local and regional meteorological conditions [6,7], the actual impact of urban emissions and those from surrounding areas on regional air quality [4,8,9], and their mutual interaction [10].

One of the major steps to properly describe the city- and regional-scale weather conditions over cities in regional scale models is the application of urban canopy (sub)models that parameterize the subscale processes that occur on street level or within buildings, but have important feedback to the model-resolvable scales [11,12]. It has been shown in many studies that the way the urban canopy and related meteorological effects are treated in models has important consequences on the concentrations of modeled species (e.g., [13–17]). This is not surprising given the fact that the transport, diffusion, deposition, and chemical transformation of pollutants (driven by the Eulerian continuity equation) are strongly dependent on meteorological conditions [18].

In particular, urban surfaces increase the air temperature, which, in turn, increases the chemical reaction rates and also the dry deposition and leads to a decrease of ozone in urban areas [13,16]. Increased temperatures in the urban canopy layer further decrease the atmospheric stability in urban areas, potentially leading to urban-breeze circulation [19–21]. The decrease of city-scale wind speeds results in limited horizontal dispersion of pollutants, which, alone, leads to elevated concentrations of primary species, especially $NO_x$ and primary aerosol [16,17,22,23]. Finally, the drag that causes the decrease of urban winds enhances turbulence, which is further increased by the buoyant forces (due to an increase of near-surface temperature), resulting in a great increase of vertical eddy diffusion. Increased eddy diffusion helps to remove species from the urban canopy and reduces the concentrations of primary species as shown by [16,17,24–27]. The consequences for ozone as a secondary species are however opposite. By removing $NO_x$ from the urban canopy by enhanced turbulence, ozone titration is reduced, leading to elevated ozone levels [26]. The large impact of the modifications of vertical turbulent diffusion on ozone levels was assessed also by Tang et al. [28]. In general, turbulence is considered to be the most important component via which the urban canopy meteorological effects act on chemistry. The overall impact, viewed as a combination of temperature, wind, and turbulence changes, is very similar to that of turbulence changes alone [29].

According to the above-mentioned facts, it is very important to analyze the sensitivity of concentrations of the modeled urban species to the numerical representation of urban land use. In two extreme cases, urban areas can either be completely ignored by the model or fully represented by an urban canopy parameterization (as done, e.g., in [16,17]), but many intermediate treatments can be considered, e.g., the BULK approach, which takes urban surfaces into account as any other flat surface with specific physical parameters (e.g., albedo), as done in [7]. The response of urban air chemistry is dependent on the urban parameterizations chosen (e.g., [24,30]) and on the settings of various parameters that describe the urban environment like the albedo of urban surfaces, their heat conductivity, the geometry of the urban setting, etc. [15].

In the Czech Republic, the capital city Prague belongs to a region with worsened air quality. This can be attributed to several factors. One important factor is the topography. The Prague basin, where the city is located, negatively affects the climatic conditions and dispersion of pollutants. During colder periods of the year, temperature inversions are often formed, leading to the accumulation of pollution. The most problematic pollutants, mainly due to heavy traffic in the city and residential heating in the suburban areas, are $NO_2$ (nitrogen dioxide), $PM_{10}$ (particulate matter of a diameter r < 10 μm) and $PM_{2.5}$ (particulate matter of a diameter r < 2.5 μm), often exceeding the limit values or leading to smog situations [31]. In particular, in January and February 2017, the threshold values of $PM_{10}$ for announcing a smog situation were exceeded three times. In the middle of January, even the criteria for announcing a regulation were fulfilled [32]. The ability of models to capture these

extreme pollution episodes correctly is especially important with regards to predicting similar events and proposing effective regulation measures.

Within the URBI PRAGENSI project supported by the Operational Programme Prague–Growth Pole of the Czech Republic (European structural and investment funds), a weather/air quality forecast system for Central Europe was set up at Charles University with high resolution nested predictions for the Czech capital Prague. These forecasts aimed to support fast action of the city council authorities if a significant worsening of air quality is predicted over the city. In order to increase the reliability and quality of the forecasts, multiple configurations have to be tested, and one of the key parts is the numerical treatment of the urbanization-induced meteorological changes (like UHI).

To fulfill this goal, we present here a study that aims to investigate to what extent the urban canopy treatment in the driving meteorological models influences the final species concentrations in the driven chemistry transport model. The basis of this study is a pair of weather prediction configurations of the WRF (Weather Research and Forecasting) Model and the consequent chemistry transport model predictions. Our focus will be a winter period with stagnant conditions, limited mixing, and very strong PM pollution.

## 2. Methods

### 2.1. Models and Data

#### 2.1.1. WRF

Meteorological variables required as inputs of the CAMx (Comprehensive Air Quality Model with Extensions) model or needed for emission pre-processing were obtained from weather forecast simulations performed with the WRF Model. WRF is a non-hydrostatic regional-scale numerical weather prediction and climate model developed to be able to simulate weather phenomena across multiple scales from the global to the city scale [33]. The WRF Model Version 4.0.3 was run in two configurations; here, the common parts are detailed. For radiation calculations, the Rapid Radiative Transfer Model for General Circulation Models (RRTMG [34]) was used. Land-surface processes were solved by the Noah land-surface model [35]. Microphysical transformations of water were treated by the improved BULK Thompson scheme [36]. For the calculation of the Planetary Boundary Layer (PBL) turbulent exchange, the Boulac PBL [37] scheme was used.

The urban canopy effects could be treated with the simple BULK approach that considers urban surfaces as flat ones with prescribed physical parameters like albedo, roughness, etc. A more comprehensive approach is offered by the multi-layer Building Environment Parameterization (BEP; [38]) linked to the Building Energy Model (BEM; [39]), the BEP+BEM method.

#### 2.1.2. CAMx

Air quality simulations were performed with the Chemistry Transport Model (CTM) CAMx Version 6.50 [40]. As an Eulerian photochemical CTM, CAMx implements multiple gas phase chemistry options (Carbon Bond chemical mechanisms: CB5, CB6; The Statewide Air Pollution Research Center chemical mechanism: SAPRC07TC). In this study, the CB5 scheme [41] was used. The static two-mode approach was used to treat particle matter physics. Dry deposition was solved using the method of Zhang et al. [42], and for wet deposition, the Seinfeld and Pandis [43] scheme was used. To calculate the composition and phase state of the ammonia-sulfate-nitrate-chloride-sodium-water inorganic aerosol system in equilibrium with gas phase precursors, the ISORROPIA thermodynamic equilibrium model was activated [44]. To compute the Secondary Organic Aerosol (SOA) chemistry, the semi-volatile equilibrium scheme SOAP (Secondary Organic Aerosol Processor) [45] was invoked.

CAMx was coupled offline to WRF, meaning that CAMx was run once WRF predictions were ready. To translate the WRF output to CAMx input fields, the WRFCAMx preprocessor which is provided along with the CAMx source code was used (see http://www.camx.com/download/

support-software.aspx). WRFCAMx takes the WRF output fields and writes them to the CAMx input format. For those CAMx input variables that were not provided in the WRF output, diagnostic methods were used. A key input for CAMx that drives the vertical transport of pollutants is the coefficient of vertical turbulent diffusion (Kv). Kv is of great importance in determining urban air pollution, and it is substantially perturbed by urban land use [29]. In this study, the method used in CMAQ (Community Multiscale Air Quality Modeling System) was adopted to calculate the Kv values. The "CMAQ" scheme [46] applies the similarity theory for different stability regimes of the boundary layer. The stability regime was determined using the dimensionless ratio of the height above the ground and the Monin–Obukhov length.

Dry deposition velocities in CAMx do not directly depend on the Kv values provided in CAMx inputs. Instead, for the aerodynamic resistance calculation (used for diffusion through the first model layer to the ground), the Louis [47] scheme was used based on the solar insolation, wind speed, surface roughness, and near-surface temperature lapse rate.

### 2.1.3. Emissions

Anthropogenic emissions corresponding to 2015, if not mentioned otherwise, were used. For the Czech Republic, the high resolution national Register of Emissions and Air Pollution Sources (REZZO—Registr emisí a zdrojů znečištění ovzduší) dataset issued by the Czech Hydrometeorological Institute (http://www.chmi.cz) was used. This inventory was completed with unreported fugitive emissions corresponding to 2017: production of coke, iron, and steel (all sources located in the Agglomeration of Ostrava/Karviná/Frýdek-Místek, near the border with Silesia), smelting works (located in the whole territory of the Czech Republic), other fugitive sources in the Agglomeration of Ostrava/Karviná/Frýdek-Místek, and fugitive emissions from quarries (whole territory of the Czech Republic). Detailed road transport emissions based on the traffic census of 2016, as well as emissions from Václav Havel Airport Prague for 2016 were prepared by ATEM (Ateliér ekologických modelů—Studio of ecological models; http://www.atem.cz).

For Poland, high-resolution emissions were provided by the authorities of GIOS (Główny Inspektorat Ochrony Środowiska) and KOBiZE (Krajowy Ośrodek Bilansowania i Zarządzania Emisjami) within the LIFE-IP MAŁOPOLSKA project (LIFE14 IPE/PL/000021; https://powietrze.malopolska.pl/en/life-project/). For Slovakia, detailed residential heating emissions and point sources that belong to Selected Nomenclature for sources of Air Pollution (SNAP) 2 were provided by the Slovak Hydrometeorological Institute. For remaining regions and sectors, the top-down CAMS (Copernicus Atmosphere Monitoring Service) European anthropogenic emissions (CAMS-REG-AP v1.1; Regional—Atmospheric Pollutants; [48]) were used. The mentioned emission sources contained annual emission totals of the main pollutants, namely nitrogen oxides ($NO_x$), sulfur dioxide ($SO_2$), carbon monoxide (CO), ammonia ($NH_3$), Non-Methane Volatile Organic Compounds (NMVOC), and particulate matter ($PM_{10}$ and $PM_{2.5}$).

The Czech REZZO and ATEM, as well as Polish and Slovak datasets were defined as area, point, and line (for road transportation) shapefiles of irregular shapes corresponding to counties, major sources (e.g., chimney stacks), and roads, while the CAMS data were provided as area and point sources on a regular grid. These raw emissions were processed using the Flexible Universal Processor for Modeling Emissions (FUME) emission model ([49]; http://fume-ep.org/). FUME is intended primarily for the preparation of emission files ready-to-use in CTMs. FUME is thus responsible for the preprocessing of the raw input files and the spatial redistribution, chemical speciation, and temporal disaggregation of input emissions. Emissions are provided in 11 SNAP activity sectors. Sector-specific NMVOC speciation was based on Passant [50] and $PM_{2.5}$ speciation on country- and sector-specific profiles provided with the CAMS data. For the vertical distribution of emissions from CAMS point sources, typical point source parameters based on the analysis of the data from the Czech database REZZO were used. Time-disaggregation factors [51,52] were applied to the spatially redistributed emissions to derive hourly emission data for CAMx. Biogenic emissions

of hydrocarbons (BVOC—Biogenic Volatile Organic Compounds) and NO were calculated based on WRF meteorological data using the MEGANv2.1 emission model [53].

### 2.1.4. Air Quality Stations: Meteorology and Pollutants

For the evaluation of the models, validated in situ measurements from air quality stations in Prague were used (Figure 1) [31]. These stations provide measurements of both air pollutant concentrations and meteorological data. In particular, $PM_{10}$ and $PM_{2.5}$ concentrations, Temperature at 2 m (T2), Wind Speed (WS), and Relative Humidity (RH) were acquired. Eight background urban and background suburban stations were considered for the air pollutants, while the meteorological observations from two available suburban stations were used. All the data were acquired from the Air Quality Information System database of the Czech Hydrometeorological Institute.

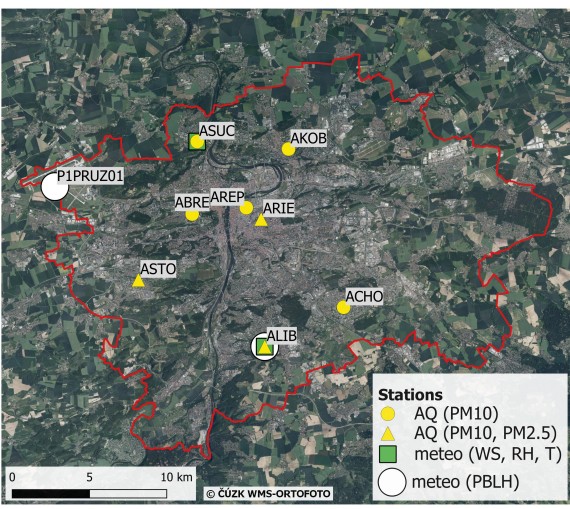

**Figure 1.** The locations of the observation sites in Prague used for evaluation of Air Quality (AQ) and meteorological (meteo) results. The labels show the code name of each station.

### 2.1.5. PBL Height Retrieval from Ceilometers

The planetary boundary layer heights (PBLHs) were retrieved from ceilometers at two suburban locations in Prague (Figure 1). The instruments used were Vaisala CL31 and CL51 ceilometers, which use pulsed diode laser LIDAR (Light Detection And Ranging) technology [54,55]. The ceilometers give three candidates for the PBLH at 16-s intervals during the entire day. Each candidate is automatically assigned a quality index, ranging from 1 (worst candidate) to 3 (best candidate; for further details, see [56]). For the purpose of this study, the best candidate for the PBL within a certain hour was determined as the candidate with the highest sum of quality indices, and its height was calculated as an arithmetic hourly mean.

### 2.2. Experimental Setup

### 2.2.1. Model Configuration

WRF (Version 4.0.3) was run on three nested domains with horizontal resolutions of 9 km, 3 km, and 1 km, with $173 \times 153$, $169 \times 151$, and $84 \times 84$ grid points, respectively (Figure 2). In the vertical dimension, the model grid had 49 levels. The height of the lowermost level was about 48–50 m (depending on the temperature of the corresponding layer). The model top was at 50 mbar. In order to assess the impact of using an advanced setup of the WRF model on CAMx simulations, when urban canopy meteorological features were calculated more comprehensively (compared to

the default "non-urbanized" case), the WRF model was run in two configurations, which differed both in the parameterizations employed and in the regime in which the forecast was performed (see below). In both regimes, the WRF output data were collected from overlapping runs of 12 h in length, taking initial and boundary conditions from the GFS operational analyses and predictions at synoptic times of 00, 06, 12, and 18 UTC [57]. During the first six hours of each run, grid nudging of global analysis was performed, and this period served as a spin-up. The hourly outputs with prediction horizons of 7–12 of each run (four segments daily) were assembled into 24 hourly outputs per day and used for the generation of inputs for CAMx. In this way, we constructed a surrogate for local analysis and tried to eliminate the drift of WRF model fields from reality. The CAMx-ready meteorological input files were then produced with the WRFCAMx preprocessor.

The two WRF configurations mentioned are:

1.  The "non-urbanized" configuration used the "BULK" (or SLAB) treatment of the urban land-surface. In the BULK treatment, the urban land-surface is regarded as any other flat surface with prescribed surface parameters (roughness, albedo, emissivity, etc.) representing zero-order effects of urban surfaces. It is clear that such an approach cannot fully resolve the 3D nature of the urban weather phenomenon (especially turbulence and radiation in street canyons [58]). The term "non-urbanized" here refers to the fact that these urban effects are largely ignored in the BULK approach. In this setup, there was no connection (restart) between any two successive 12 h runs described above, i.e., WRF adopted the so-called cold start concept [59].

2.  The "urbanized" configuration of WRF had a more comprehensive treatment of the urban canopy. Instead of the BULK approach, the BEP+BEM urban canopy model was used (cf. 2.1.1). Moreover, significant changes to the land use data were made, resulting in more realistic values of variables describing the urban geometry and physical properties of surfaces (roofs, roads, walls), which influence the exchange between the urban canopy and the atmosphere. Since the state of the urban canopy submodel evolves in time, it was necessary to keep the continuity of its evolution throughout the simulation. Therefore, instead of a cold start as above, a restart was performed each 6 hours with the help of WRF's restart capability. The restart files produced at the end of a run, needed for the restart of a successive simulation, were enriched with urban variables. Thus, the WRF run mimicked a longer term simulation, driven by analyses and keeping the continuous evolution of the variables that describe the physical state of the urban environment.

For both configurations, land cover information was taken from the high resolution (100 m) CORINE Land Cover (CLC) 2012 data (https://land.copernicus.eu/pan-european/corine-land-cover). For the Prague urban area, the values of the urban fraction and other urban parameters used by BEP+BEM were adopted from multiple high resolution data sources. Three land cover sources were combined into one spatially-resolved dataset: the Urban Atlas 2018 (Copernicus Land Monitoring Services, Urban Atlas; https://land.copernicus.eu/local/urban-atlas/urban-atlas-2018?tab=metadata), the ZABAGED geodatabase (Základní báze geografických dat České republiky—The Fundamental Base of Geographic Data of the Czech Republic; The Czech Office for Surveying, Mapping and Cadastre; https://geoportal.cuzk.cz/), and the digital technical map of Prague provided by GeoPortal Prague (https://www.geoportalpraha.cz/en). Further, land cover fractions without natural vegetation were prepared using the European Settlement Map (Release 2017; Copernicus Land Monitoring Services; https://land.copernicus.eu/pan-european/GHSL/european-settlement-map/esm-2012-release-2017-urban-green?tab=metadata). For detailed building information, the 3D model of Prague was used (3D model; GeoPortal Prague; https://www.geoportalpraha.cz/en), and information about street geometry was based on ZABAGED. Vector data provided by the listed sources were aggregated to gridded data with 333 m spatial resolution. Emissivities of different urban surfaces were based on the ASTER Global Emissivity Dataset with 100 m spatial resolution (ASTER GED; https://emissivity.jpl.nasa.gov/aster-ged) and the Land Surface Emissivity algorithm for LANDSAT-8 [60]. In summary, urban parameters were considered as 2D arrays instead of constant

values, i.e., every urban grid point had a unique combination of urban parameters allowing capturing the spatial distribution of urban effects more precisely. An example of street width and vegetation fraction at the original 333 m × 333 m resolution is provided in Figure 3.

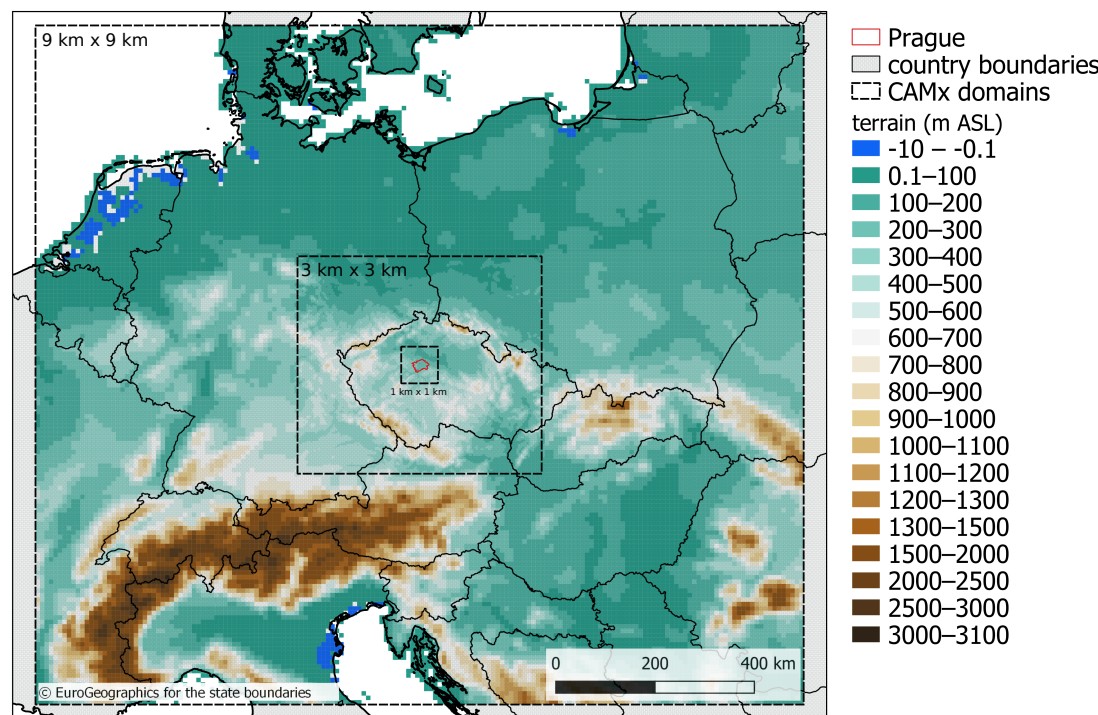

**Figure 2.** The simulation domains of WRF and the Comprehensive Air Quality Model with Extensions (CAMx).

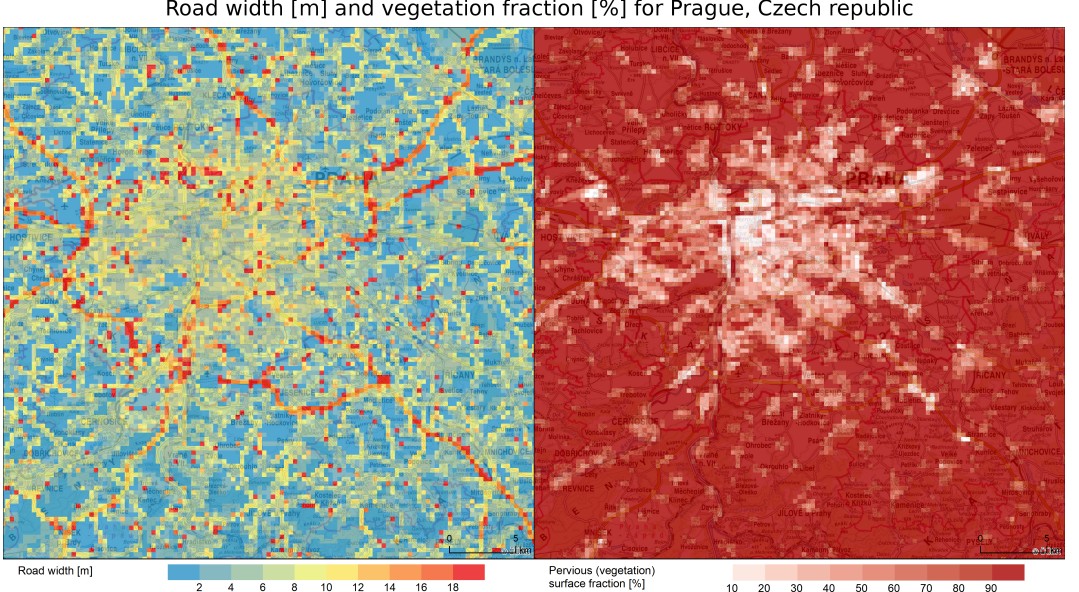

**Figure 3.** The final road width (meters) and vegetation fraction (%) map at the original 333m resolution used to derive the 2D urban input for Prague.

In this validation study, CAMx prediction was performed for each day from 00 UTC to 24 UTC and was restarted from the run of the previous day (using the CAMx restart files for 00 UTC of the actual day). As a first step of the CTM run, the Chemical Boundary Conditions (CHBC) were calculated; in this preliminary setup, they were taken as time- and space-invariant climatological means, so only the effects of the local emissions were taken into account without considering long-range pollution transport. Simultaneously with CHBC, the photolysis rate files were created with the TUV (Tropospheric Ultraviolet-Visible [61]) Model, which were based on Total Ozone Mapping Spectrometer (TOMS) data of the Ozone Monitoring Instrument (OMI) aboard NASA's Aura satellite https://earthdata.nasa.gov/eosdis/science-system-description/eosdis-components/lance/about-omi-sips. Besides the boundary conditions and TUV files, the biogenic emissions of isoprene, monoterpenes, and other volatile organic compounds were calculated based on the WRF predicted meteorological data. The start of the CAMx run was conditioned upon the successful execution of all the mentioned preprocessing steps.

The model configuration for both WRF and CAMx runs is summarized in Table 1. Both WRF and CAMx simulations were performed on an Intel Xeon-based HPC (High Performance Computing) cluster at the Faculty of Mathematics and Physics (Charles University, Prague), with parallelization using the OpenMPI (Message Passing Interface) over OmniPath technology and GNU compilers.

**Table 1.** Model configuration for WRF (left) and CAMx (right). CMAQ, Community Multiscale Air Quality Modeling System; RRTMG, Rapid Radiative Transfer Model for General Circulation Models; TUV, Tropospheric Ultraviolet-Visible Model; CB5, Carbon Bond chemical mechanism; SOAP, Secondary Organic Aerosol Processor; BEP + BEM, Building Environment Parameterization linked to Building Energy Model.

| WRF | CAMx |
|:---:|:---:|
| Geographic Projection | |
| Lambert Conformal Conic, Center: 50.075 N 14.44 E, True Latitudes: 50.075 N/50.075 N | |
| Domains (centered on the projection center) | |
| 173 × 153 (9 km); 169 × 151 (3 km); 84 × 84 (1 km) as grid points | 172 × 152 (9 km); 164 × 146 (3 km); 74 × 74 (1 km) as grid boxes |
| 49 vertical levels (up to 50 mbar) | 20 vertical layers (up to 12 km) |
| PBL parameterization/vertical diffusivities' calculation | |
| Boulac PBL [37] | CMAQ scheme [46] |
| Microphysics scheme/wet deposition | |
| Thompson scheme [36] | Seinfeld scheme [43] |
| Land surface processes/dry deposition | |
| Noah [35] | Zhang scheme [42] |
| Radiation/UV-photolysis | |
| RRTMG [34] | TUV [61] |
| Gas phase chemistry | |
| - | CB5 [41] |
| Inorganic aerosol chemistry | |
| - | ISORROPIA [44] |
| Organic aerosol chemistry | |
| - | SOAP [45] |
| Urban canopy model | |
| BULK/BEP + BEM[38,39] | - |

### 2.2.2. Simulated Period

Our period of interest was 11 January to 20 February 2017. Both January and February were characterized by worsened dispersion conditions. The period was chosen to cover two significant smog episodes with very high PM concentrations largely exceeding the limit values, which occurred 19–24 January and 14–17 February (Figure 4). From the middle of January, the synoptic situation was dominated by anticyclonal weather together with very low temperatures, going down to $-18\,°C$. Since 18 January, the sky was clear, and these conditions led to the formation of a very strong temperature inversion. Based on measurements at Prague-Libuš station (balloon soundings), the inversion layer

reached up to 1 km in height during the period 20–22 January. Together with significantly reduced wind speeds, there was very limited mixing, and these factors caused extensive accumulation of pollutants at the ground level. $PM_{10}$ concentrations exceeded the daily limit value (50 $\mu g\ m^{-3}$) at all eight Prague background stations. The maximum daily concentrations were reached on 20–21 January when all stations measured over 150 $\mu g\ m^{-3}$ and up to 180 $\mu g\ m^{-3}$. Maximum hourly concentrations were 359 $\mu g\ m^{-3}$. The maximum daily value for $PM_{2.5}$ was 147 $\mu g\ m^{-3}$, and the maximum hourly value was 203 $\mu g\ m^{-3}$. During 23–24 January, the Czech Republic was overcast by low clouds, leading to a smaller drop in temperature and improvement of dispersion conditions.

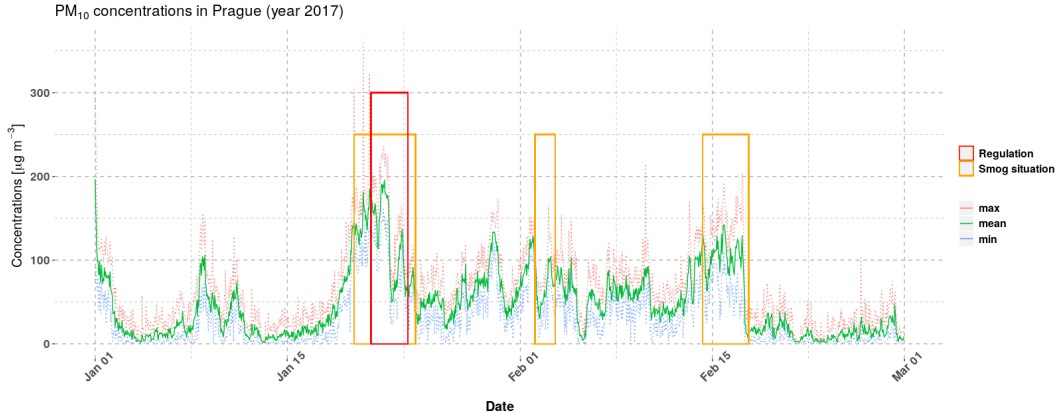

**Figure 4.** Minimum, mean, and maximum hourly $PM_{10}$ concentrations from eight background stations in Prague with the indication of the announced smog situations and regulation.

At the beginning of February, several frontal systems passed over the Czech Republic causing a short-term reduction of PM concentrations. In the middle of the month, the weather was influenced by a strong anticyclone, which led to worsening of the dispersion conditions and an increase of PM concentrations. The inversion layer reached several hundreds of meters and was most pronounced from 14 to 16 February. Although the second episode was less severe than the first one, the limit value for $PM_{10}$ was also exceeded at all eight background stations, and six of them measured over 100 $\mu g\ m^{-3}$ on 15–16 February. The maximum hourly value was 215 $\mu g\ m^{-3}$. For $PM_{2.5}$, the maximum measured daily value was 98 $\mu g\ m^{-3}$, and the maximum hourly value reached 165 $\mu g\ m^{-3}$. This episode ended during the night of 17 February when a cold frontal system passed over the Czech Republic. A detailed characterization of the conditions is described in [62].

## 3. Results

The results of the WRF and CAMx simulations with the urbanized (BEP+BEM) scheme and the non-urbanized (BULK) configuration for the period from 11 January to 20 February were compared to observations in order to evaluate the performance of the models and the differences arising from different urban canopy treatment. In this paper, we present the comparison only for the innermost domain with 1 km resolution, as the impact of the urban parameterization turned out to be more pronounced than the change of resolution. We analyzed the spatial differences, as well as the overall agreement with measurements. Hourly data and averaged diurnal profiles of the meteorological conditions and pollutants at the points of measurement have been assessed. For the WRF model, the closest grid point to each measurement site was used, and for CAMx, the values from the grid box that lied above the measurement station were extracted for comparison.

### 3.1. Meteorology

Table 2 summarizes the values of the bias, the Root Mean Square Error (RMSE), and the correlation coefficient *r* (see Appendix A) together with the correlation *p*-value for the comparison of meteorological variables with observational data from two stations in Prague. The diurnal profiles for

the modeled and observed air Temperature at 2 m (T2), Wind Speed (WS), Relative Humidity (RH), and Planetary Boundary Layer Height (PBLH) averaged over the modeled period are shown in Figure 5. It can be seen that for T2, the bias was significantly reduced in the urbanized configuration as compared to the BULK configuration. The RMSE was the same in both cases, while the correlation was poorer. As expected, the BULK configuration gave more underestimated temperatures, on some occasions up to 10 °C lower than those predicted with the urbanized configuration. From the averaged diurnal profiles, it could be observed that the daily pattern was captured quite well by the urbanized model, with a small overestimation during the night time. The BULK scheme gave values systematically lower by 2 °C throughout the day.

**Table 2.** Statistical results (bias, RMSE, $r$, correlation $p$-value) of observations and simulations with corresponding units for bias and RMSE: meteorology; hourly averages. T2, Temperature at 2 m; WS, Wind Speed.

| Scheme | | T2 (°C) | WS (m s$^{-1}$) | RH (%) | PBLH (m) |
|--------|--------------|---------|------------------|--------|----------|
| BULK | bias | −1.9 | 1.5 | 3.3 | −229 |
| | RMSE | 2.7 | 2.4 | 11.5 | 369 |
| | $r$ | 0.91 | 0.79 | 0.34 | 0.5 |
| | $p$-value | <0.01 | <0.01 | <0.01 | <0.01 |
| BEP+BEM | bias | 0.6 | 1.3 | −3.2 | −134 |
| | RMSE | 2.7 | 2 | 11.4 | 322 |
| | $r$ | 0.82 | 0.75 | 0.45 | 0.5 |
| | $p$-value | <0.01 | <0.01 | <0.01 | <0.01 |

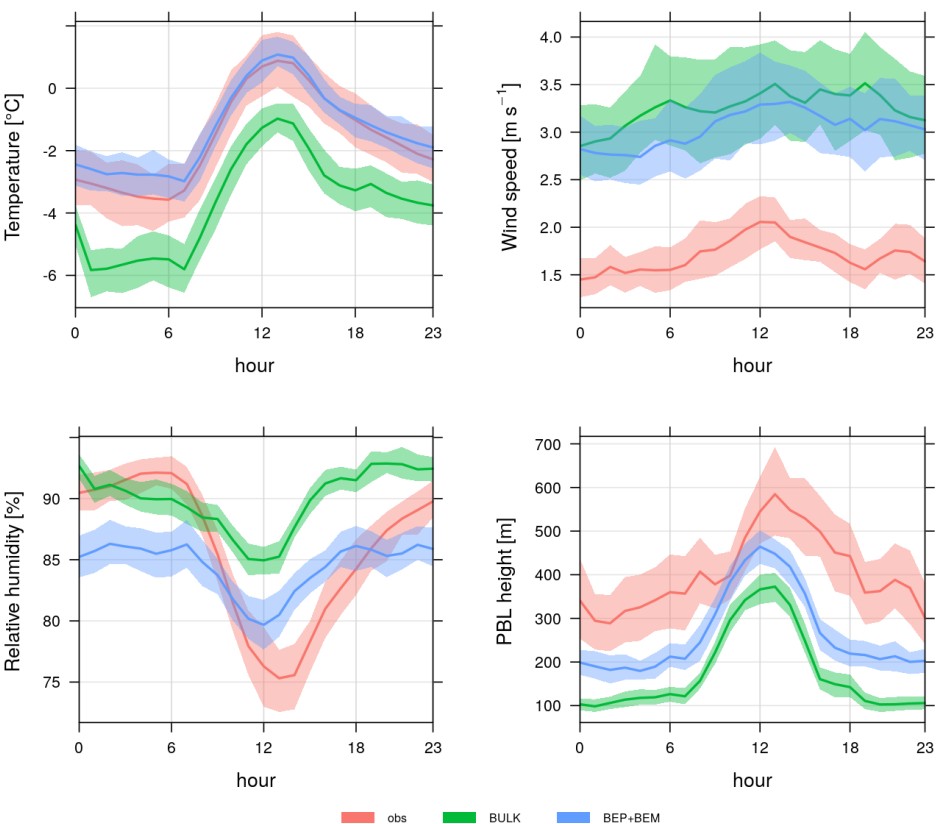

**Figure 5.** Averaged diurnal profiles of the temperature at 2 m, wind speed, relative humidity, and PBL height for observations (red), BULK simulations (green), and BEP+BEM simulations (blue). Lines represent the mean values and shaded areas the 95% confidence interval in the mean.

Similar results could be observed for wind speed. In this case, the results of both runs were overestimated, the non-urbanized predictions being up to 6 m s$^{-1}$ (0.5 m s$^{-1}$ on average) higher than the urbanized ones, which were closer to the observed values. RMSE was better for the BEP+BEM configuration; however, the correlation was somewhat lower in this case.

The relative humidity had a much larger diurnal amplitude in the observed data (almost 17% from 75 to 92%) with an expected maximum in the morning hours and a minimum just after noon. This pattern failed to be captured by either of the configurations: the daily amplitude was slightly over 5% with the urbanized model giving values constantly lower by also about 5%. In both configurations, the biases (in an absolute sense) and RMSE were very similar; however, the correlation for the urbanized version was somewhat higher.

The simulated planetary boundary layer height using the BEP+BEM scheme was improved in terms of bias and RMSE, while both configurations were marked with negative bias compared to measurements. The daily averaged observed heights ranged from 300 m during the night to 600 m around noon; the configuration without urban parameterization predicted heights between 100 m and 400 m; and the urbanized model heights ranged from 200 m to 500 m. The correlation between the two versions and the measurements was almost the same.

A paired *t*-test was performed for all variables, and in all cases, a significant difference between the BULK and BEP+BEM scenarios at a confidence level of 99% was confirmed.

### 3.2. Air Quality

### 3.2.1. PM$_{2.5}$ and PM$_{10}$

As said, the impact of the model treatment of urbanized surfaces propagated to the impact on near-surface PM$_{2.5}$ and PM$_{10}$ concentrations through modifications in the meteorological conditions driving the transport, diffusion, and chemical transformations of pollutants. It was therefore expected that the differences in meteorological conditions presented above would imply differences in pollutant concentrations. Here, we compare these two configurations in terms of the two aerosol fractions; both their spatial distribution and the agreement with station data are evaluated.

The statistics applied for the comparison of both model setups to observations are summarized in Table 3. Further, we calculated the diurnal profiles averaged over the modeled period (Figure 6). For both pollutants, the bias and RMSE were higher for the urbanized configuration. The correlation for the urbanized version was somewhat lower as well; however, the averaged diurnal profiles showed that the temporal patterns of the urbanized configuration were better correlated with the measurements than the BULK runs, although more underestimated. In particular, the correlation coefficient of the averaged diurnal cycles increased from 0.06 to 0.3 for PM$_{10}$ and from 0.55 to 0.69 for PM$_{2.5}$. This could indicate that the day-to-day variability was better captured in the non-urbanized version (unlike the average hourly one, which was better in the urbanized version). Both model outputs showed greater variations during the day than the observed data, especially with a higher peak in the afternoon, which was not present in the measurements. In particular, the average measured concentrations of PM$_{10}$ varied from 50–65 µg m$^{-3}$; the BULK simulated values ranged from 45–80 µg m$^{-3}$ with two distinct peaks in the morning and evening rush hours; and the BEP+BEM model outputs varied from 30–50 µg m$^{-3}$, also with two peaks. For PM$_{2.5}$, the concentrations were around 40–55 µg m$^{-3}$, 35–55 µg m$^{-3}$, and 25–40 µg m$^{-3}$, respectively.

The day-to-day variation of the daily average PM$_{2.5}$ and PM$_{10}$ concentrations (Figure 7) confirmed the diurnal average picture: i.e., systematically lower concentrations in the BEP+BEM case seen on every simulated day; and the differences in both cases could reach 40–50 µg m$^{-3}$, especially for PM$_{10}$. Another important conclusion was that during days with low measured PM concentrations, the BEP+BEM version had a smaller bias, or in other words, the negative bias in the BEM+BEP case was due to large overestimation of the peak measured values (e.g., during the beginning of the examined period).

Figure 8 shows the surface concentrations of $PM_{10}$ and $PM_{2.5}$ averaged over the simulated period for the BULK (non-urbanized) and BEP+BEM (urbanized) configurations. A clear decrease of modeled $PM_{10}$ concentrations could be observed in the urbanized simulation, not only over the urban areas, but also over a major part of the domain, in agreement with the diurnal cycles and daily averages. The average concentration difference over the area of Prague between the BULK and BEP+BEM results was 12 µg m$^{-3}$. Similar results were obtained for $PM_{2.5}$, where the mean decrease of concentrations in Prague reached 8 µg m$^{-3}$. There was also a visible difference in spatial patterns. In the BULK configuration, the highest $PM_{10}$ values were simulated over the urban areas of Prague with a hotspot in the inner part of the city. Elevated concentrations could also be observed in other urban areas north of Prague. However, the simulations with the BEP+BEM configuration gave more evenly mixed concentrations and even a decrease towards the center of the city. The spatial variability of the modeled $PM_{2.5}$ concentrations was smaller than that of $PM_{10}$. Similarly to the $PM_{10}$ results, the non-urbanized concentration predictions were highest in the central part of the city. The urbanized configuration led to an overall decrease in concentrations over the entire domain, especially in the urban area of Prague. A paired *t*-test was performed for both configurations, and in all cases, a significant difference between the BULK and BEP+BEM scenarios at a confidence level of 99% was confirmed.

**Table 3.** Statistical results (bias, RMSE, *r*, correlation *p*-value) of observations and simulations: air quality; hourly and daily averages. Bias and RMSE units are in µg m$^{-3}$.

| Scheme | | $PM_{10}$ | $PM_{2.5}$ | $PM_{10}$ Diurnal | $PM_{2.5}$ Diurnal |
|---|---|---|---|---|---|
| BULK | bias | 0.06 | −6.7 | −1 | −6.6 |
| hourly | RMSE | 36.4 | 28.7 | 12.69 | 8.9 |
| | *r* | 0.62 | 0.66 | 0.06 | 0.55 |
| | *p*-value | <0.01 | <0.01 | <0.01 | <0.01 |
| BEP+BEM | bias | −20.1 | −18.3 | −20.2 | −18.3 |
| hourly | RMSE | 41 | 35.8 | 21.1 | 18.7 |
| | *r* | 0.59 | 0.58 | 0.3 | 0.69 |
| | *p*-value | <0.01 | <0.01 | <0.01 | <0.01 |
| BULK | bias | 0.5 | −6.3 | | |
| daily | RMSE | 23.1 | 20.9 | | |
| | *r* | 0.8 | 0.8 | | |
| | *p*-value | <0.01 | <0.01 | | |
| BEP+BEM | bias | −19.7 | −18 | | |
| daily | RMSE | 34.5 | 30.5 | | |
| | *r* | 0.7 | 0.7 | | |
| | *p*-value | <0.01 | <0.01 | | |

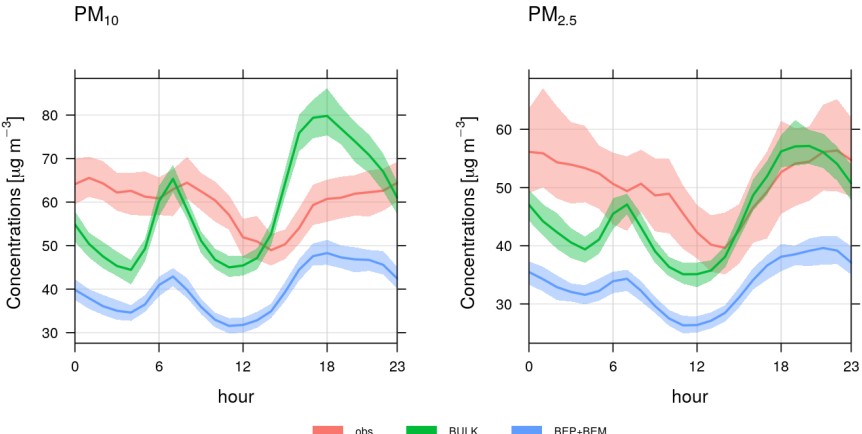

**Figure 6.** Averaged diurnal profiles of $PM_{10}$ and $PM_{2.5}$ concentrations for observations (red), BULK simulations (green), and BEP+BEM simulations (blue). Lines represent the mean values and shaded areas the 95% confidence interval in the mean.

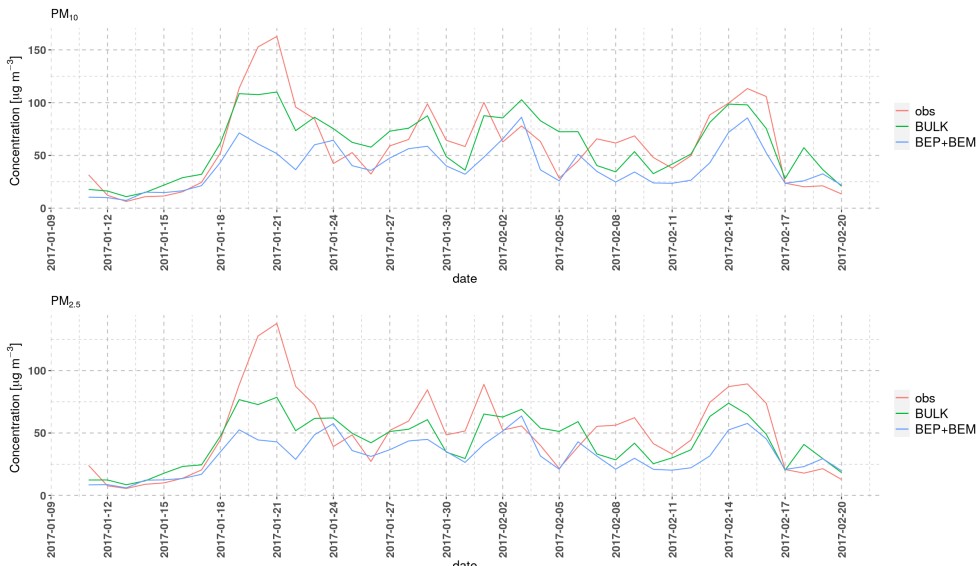

**Figure 7.** Average daily concentrations of $PM_{10}$ (top) and $PM_{2.5}$ (bottom) for observations (red) and the BULK (green) and BEP+BEM (blue) model configurations.

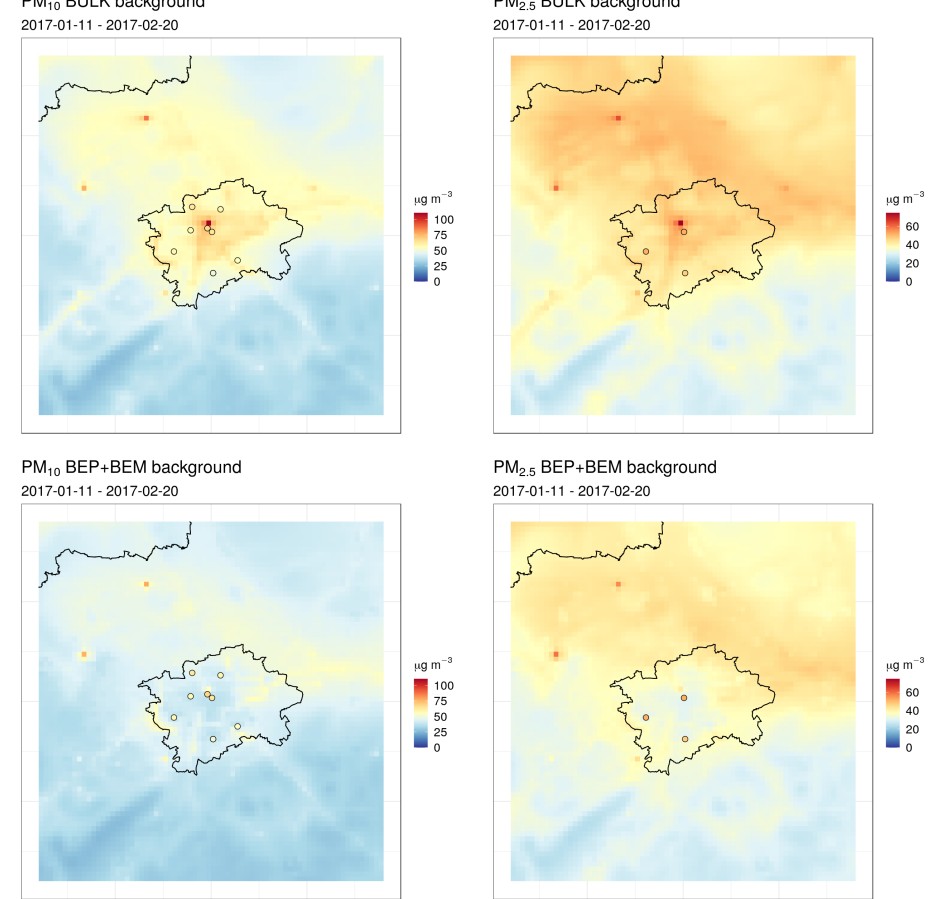

**Figure 8.** Maps of modeled $PM_{10}$ (left) and $PM_{2.5}$ (right) concentrations averaged over the entire period with the BULK scheme (top) and BEP+BEM scheme (bottom). Circles represent measurement stations and their average concentrations from the same period.

### 3.2.2. Analysis of the Aerosol Components

Huszar et al. [17] showed that different aerosol components contribute to the total urbanization-induced $PM_{2.5}$ change with a different magnitude. In order to assess this in our simulations, we analyzed the differences in concentrations of both primary and secondary aerosol between the two configurations to evaluate the impact of the urbanized version of the forecast (with respect to the "BULK" approach). The diurnal cycle of the Primary Aerosol (PA) concentrations from the two model versions is presented in Figure 9 and the corresponding day-to-day variations of the average daily values is shown in Figure 10. Figure 11 then presents the case of Secondary Inorganic Aerosol (SIA), nitrates ($PNO_3$), sulfates ($PSO_4$), and ammonium ($PNH_4$) with the respective daily averages in Figure 12. The diurnal cycle for the Secondary Organic Aerosol (SOA) is plotted in Figure 13 and the corresponding day-to-day variations are presented in Figure 14.

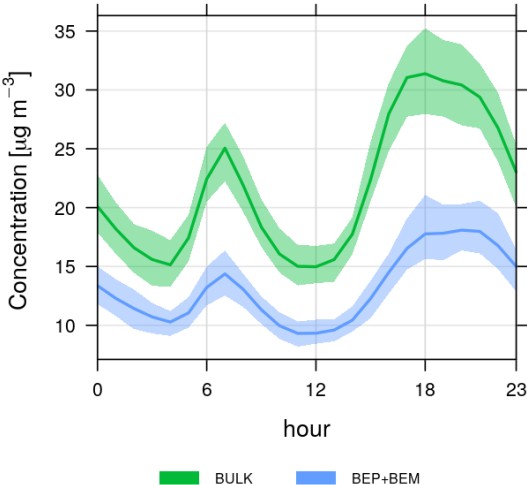

**Figure 9.** Averaged diurnal profiles of PM primary aerosol for BULK simulations (green) and BEP+BEM simulations (blue). Lines represent the mean values and shaded areas the 95% confidence interval in the mean.

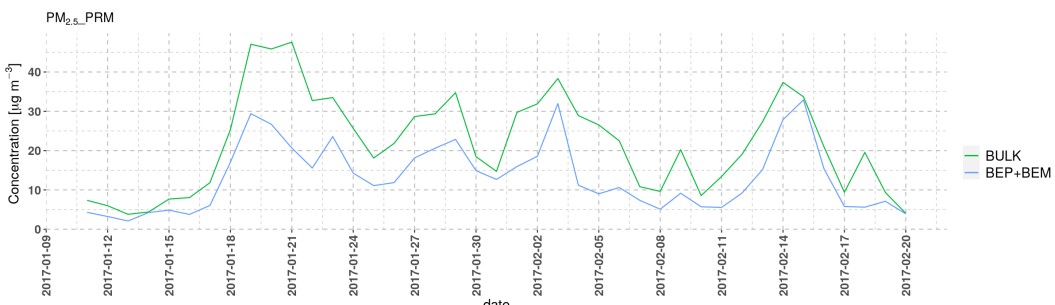

**Figure 10.** Average daily concentrations of PM primary aerosol for BULK simulations (green) and BEP+BEM simulations (blue).

According to Figure 9, primary aerosols exhibited a very similar diurnal cycle to the total $PM_{2.5}$, comprising about 50–60% of the total aerosol load. The BEP+BEM version showed lower PA by about 15–25 µg m$^{-3}$ compared to the BULK version, and this behavior was systematically seen during every simulated day. This indicated that about half of the difference seen in Figure 6 between the two versions was caused by the differences in primary aerosol.

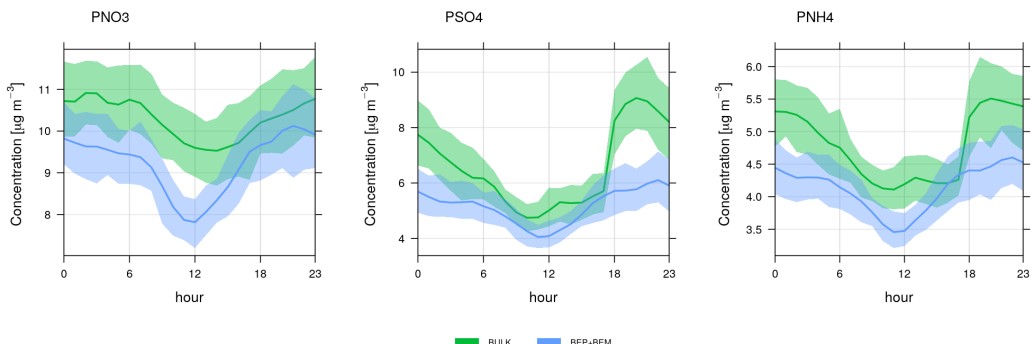

**Figure 11.** Averaged diurnal profiles of PM secondary inorganic aerosol (nitrates-PNO$_3$, sulfates-PSO$_4$, and ammonium-PNH$_4$) for BULK simulations (green) and BEP+BEM simulations (blue). Lines represent the mean values and shaded areas the 95% confidence interval in the mean.

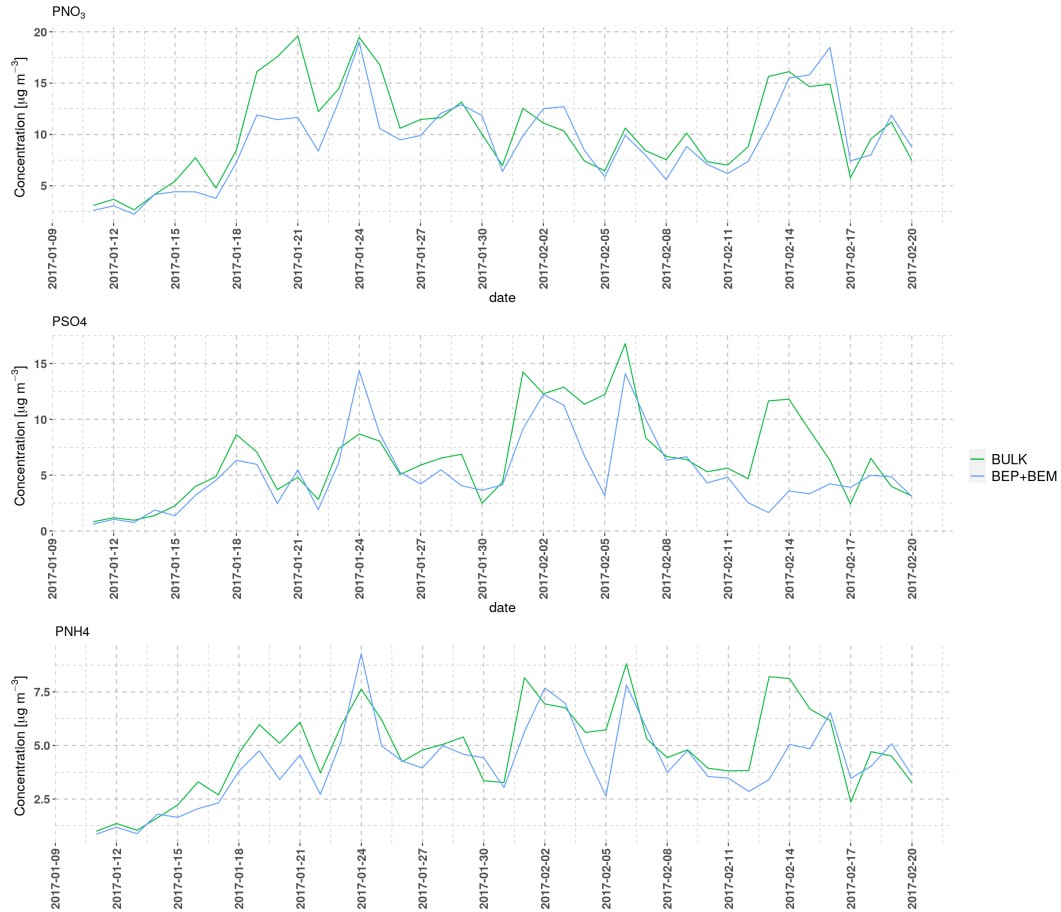

**Figure 12.** Average daily concentrations of PM secondary inorganic aerosol (nitrates-PNO$_3$, sulfates-PSO$_4$, and ammonium-PNH$_4$) for BULK simulations (green) and BEP+BEM simulations (blue).

Regarding secondary inorganic aerosol (Figure 11), in all cases, the BEP+BEM version calculated smaller values; however, the differences were much lower than in the case of PA. For nitrates and sulfates, it showed about 1–2 µg m$^{-3}$ and for ammonium about 1 µg m$^{-3}$. The diurnal cycle of these components showed a different pattern compared to the primary aerosol with maximum values during the nighttime. The differences were however largest during the daytime for nitrates, while for sulfates and ammonium, the difference peaked rather during the nighttime. The differences were however not uniform during individual days (Figure 12), and under certain conditions, the BEP+BEM configuration could lead to higher aerosol loads. This was the case for 24 January, which ended

the strong inversion period, and the turbulence from the upper layer above the original inversion layer could become of greater importance. Indeed, the turbulence was stronger in the BEP+BEM case, which could explain the slightly higher concentrations. A similar situation was modeled after the second strong inversion period, which ended on 16 February. Here, turbulent transport from upper layers could explain the slightly higher SIA values in the BEP+BEM configuration.

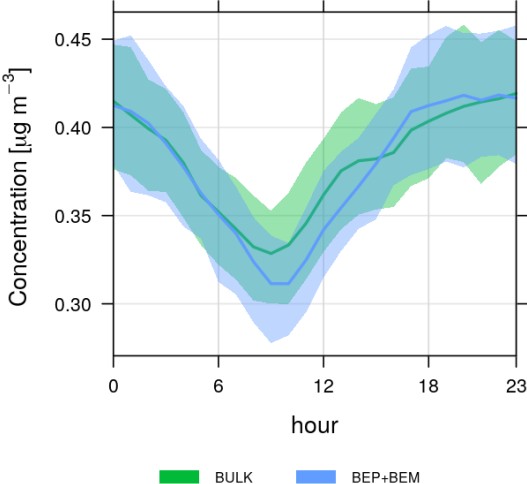

**Figure 13.** Averaged diurnal profiles of PM secondary organic aerosol for BULK simulations (green) and BEP+BEM simulations (blue). Lines represent the mean values and shaded areas the 95% confidence interval in the mean.

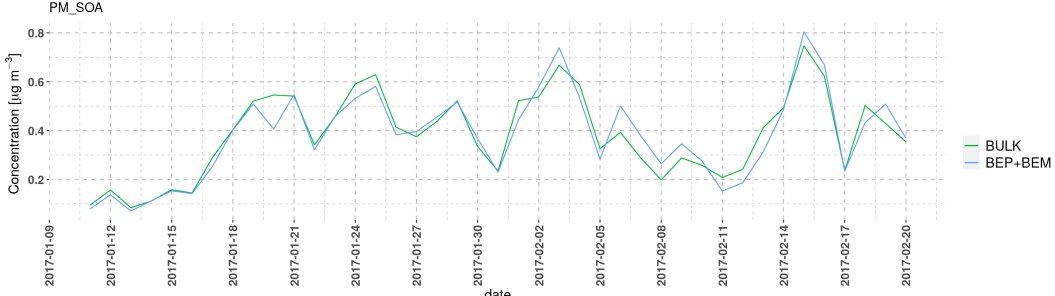

**Figure 14.** Average daily concentrations of PM secondary organic aerosol for BULK simulations (green) and BEP+BEM simulations (blue).

SOA comprised only a small fraction of the total $PM_{2.5}$ in our simulations, as seen in Figure 13. The diurnal cycle was similar to that of the SIA with nighttime maxima, while the differences between the two model versions were negligible, less than 0.05 µg m$^{-3}$. This conclusion remained the same also from the day-to-day figure. In summary, the modeled $PM_{2.5}$ diurnal cycle was determined mainly by the primary aerosol. The same conclusion could be made for the differences between the BEP+BEM and BULK versions.

## 4. Discussion

The presented differences between the non-urbanized and urbanized version of the WRF setup showed expected patterns: higher temperatures in the urbanized version as a consequence of accounting for the urban meteorological effects such as radiation trapping and anthropogenic heat release [63]. Indeed, the "BULK" approach could not resolve these effects. Liao et al. [24] also simulated higher temperatures using WRF with the BEP+BEM configuration in comparison with their SLAB scheme (which corresponds to our BULK approach). They however detected smaller

differences between these two approaches. The reason could lie in the fact that in our case, the WRF forecasts with the BULK treatment were cold-started, being systematically closer to the global driving data (GFS), which were apparently negatively biased for urban areas. In the case of [24], the correlations were however slightly worse for the urbanized version, which could suggest that the urban morphology parameters were not entirely realistic in their study. Sharma et al. [58] simulated a smaller temperature bias when using the BEP+BEM instead of the SLAB scheme, especially for nighttime hours. Higher temperatures were simulated for the BEP+BEM compared to the BULK scheme in [7] as well (as a ten-year average); here, however, the differences were smaller, again probably for the same reasons as noted above. Similarly, Wang et al. [30] compared the BULK and SLUCM (Single-Layer Urban Canopy Models) in WRF. SLUCM is a simpler, single-layer approach to describing the urban environment compared to the multilayer BEP+BEM approach presented here. In their study, the SLUCM captured temperature with a smaller bias and RMSE compared to the BULK approach. However, interestingly, their correlation was lower for the more comprehensive approach of SLUCM. This was also true for our case (and also in [24]) and could suggest that the high-frequency variability (hour-by-hour) was less accurately captured with a more advanced urban canopy description as it was always marked with higher number of degrees of freedom (note the large number of urban parameters that needed to be set). This also stressed the need for very careful setting of urban parameters as they have a very large impact on model results and can potentially lead to an increase in model errors.

Regarding wind speed, it was clear that the BEP+BEM configuration accounted for the increased and complex three-dimensional nature of drag in the urban canopy in comparison to the BULK approach, where the urban drag was imposed only by an increased roughness of the surface. This explained why the winds were lower in the BEP+BEM case. This was supported by the findings of Karlicky et al. [7], who simulated long-term winter wind speeds lower by 2 m s$^{-1}$, a larger decrease than in our case. Lower wind speeds were also simulated in [30] by around 0.5–1 m s$^{-1}$, which was very similar to our result. Improvement in wind speed bias due to the application of an urban canopy model was achieved in [24,58] as well. However, in our case and in all of the mentioned studies, the correlation was higher for the BULK approach, and the reasons were probably similar to the temperature case.

Lower relative humidities in case of the BEP+BEM simulations were clearly a result of higher temperatures, as evaporation of water vapor from the ground was the same in both cases (in the BULK case, surfaces were of urban land use type, so evaporation was also limited). Lower RHs were modeled by [30] as well. It has to be noted however that the main reason for smaller relative humidities in urban areas lies in the limited evaporation and high run-off, as argued by [64].

The increase of the urban PBLH when switching from the BULK approach to BEP+BEM was attributable to better representation and thus stronger turbulence in the urban boundary layer. This was concluded by [24] as well, who simulated higher PBLH by 200–300 m in BEP+BEM simulation compared to the BULK approach. Their differences were larger than our 100 m, but the urban agglomeration they analyzed was much larger with stronger potential impacts. Similarly to our results, Wang et al. [30] simulated improvements in bias and RMSE when applying a more sophisticated urban canopy treatment instead of the BULK approach. About 100 m higher PBLH was modeled in [7,23], using BEP+BEM (BEP in the case of the first study) instead of the BULK approach. The Boulac PBL was used recently by [65] as well, who concluded, in accordance with our results, that urban PBLs were higher and thus much more accurately represented in a multi-layer urban canopy scheme (as BEP) compared to a single-layer approach.

The simulated PM$_{10}$ and PM$_{2.5}$ concentrations were systematically larger in the BULK case. This was very much in line with the expectations regarding the vertical mixing in the two versions. In BEP+BEM, the PBLH was higher, indicating stronger turbulence and hence larger turbulent transport and removal of pollutants. Huszar et al. [17,29] also showed that turbulence was a primary factor that played a role in the urban-induced meteorological modifications in cities. Liao et al. [24] also showed that the BEP+BEM scheme resulted in lower PM$_{10}$ concentrations by about 15–20 μg m$^{-3}$.

Although wind speeds were lower in the BEP+BEM scheme in our simulation, the dominance of the enhanced turbulence in this model version was confirmed [29].

Our results were also in line with Kim et al. [26], who simulated lower $PM_{2.5}$ concentrations when using a more comprehensive treatment of the urban canopy than a simple BULK approach with differences reaching about 5–10 $\mu g\ m^{-3}$. Similar decreases for $PM_{10}$ were modeled by Zhu et al. [66], who analyzed the effect of urban expansion, which resulted in increased vertical eddy mixing and therefore lower near-surface pollutant concentrations.

Our PM biases for the BEP+BEM case and the difference between the BEP+BEM and BULK case were larger during peak measured values, and on the other hand, during low PM values, the BEP+BEM tended to perform with a smaller bias. This strongly suggested that the turbulence was somewhat overestimated in BEP+BEM during very stable conditions, while it performed more realistically in other situations.

It must be stressed that although in most of the studies, turbulence had a dominant effect, under certain conditions, wind speed could play a major role as well and change the overall picture regarding the impacts of urban areas. For example, de la Paz et al. [23] calculated higher near-surface concentrations for $PM_{2.5}$ when using the BEP (without BEM) approach in WRF instead of the BULK one and concluded the dominant role of decreased wind speeds in urban areas preventing dispersion of primary emissions. In their case, the urbanized approach yielded a much larger difference in wind speed (up to 2 $m\ s^{-1}$) between the two setups than in our study. Our results showed that if turbulence became dominant, the conclusions could be qualitatively different.

There were striking differences in the shape of the PM diurnal cycles between modeled and observed values, especially in the case of $PM_{10}$. As PBL and wind were simulated with much lower bias and higher correlation, we concluded that the main reason lied in an incorrect representation of the diurnal cycle of the emissions. Our diurnal emission factors were taken from Gon et al. [52], and probably their hourly distribution of vehicle emissions overestimates the rush hour peaks that are not seen in the measured data from Prague.

Our analysis showed that the modeled differences of $PM_{2.5}$ concentrations between the two versions were well explained by the primary component of $PM_{2.5}$ (primary aerosol), which had an almost identical shape in diurnal variation. Indeed, Huszar et al. [17] showed that the urban induced changes in $PM_{2.5}$ were largely determined by the primary aerosol and that the main driver of the primary aerosol decrease was the elevated turbulence that removed material from the urban canopy layer to the mixing layer above. Our largest differences in PA were modeled during early evening hours, which were very in line with the results of [17,29], who showed that turbulence caused the largest impact (decrease) during this period of the day.

The modeled differences for secondary inorganic aerosol had, besides increased turbulence, another cause: the differences in temperatures between the two model versions. Temperature acted as a major driver for secondary aerosol gas-particle partitioning, with increasing temperatures resulting in less secondary aerosol forming [67]. The largest impact of the temperature itself was modeled in [17] during evening hours. Indeed, during these hours, the impact on temperature was the largest in our simulations, and this probably translated to the large impact of the BEP+BEM on SIA during nighttime, especially for ammonium and sulfates.

Apparently, our $PM_{2.5}$ model performance was worse for the urbanized version as this decreased the $PM_{2.5}$ values, making the negative bias even larger. This however did not mean that urbanization of the prediction system was the wrong step towards a more accurate weather/air-quality forecast system. This meant (and confirmed) that the overall model bias had many sources that caused the model results to deviate in both directions with respect to the observed values. It was likely that the emissions of PM were too low, creating the general negative bias, which was intensified with the introduction of the urbanized model version with BEP+BEM.

## 5. Conclusions

A high resolution weather and air-quality forecast system was set up using the WRF numerical prediction model and CAMx chemistry transport model for the city of Prague, Czech Republic. This system was used to simulate a typical winter pollution episode in January–February 2017 using two distinct setups: BULK configuration using the "BULK" treatment of urban land-surface and the BEP+BEM (Building Environment parameterization linked to Building Energy Model) configuration, taking the urban canopy into account. The effects of the urbanization were compared to the non-urbanized simulations, and both outputs were evaluated against observations.

The results showed that the urbanized runs were able to capture the meteorological conditions better than the non-urbanized simulations. In particular, we observed an average increase in temperature at 2 m by approximately 2 $^\circ$C, a decrease in wind speed by around 0.5 m s$^{-1}$, a decrease in relative humidity by 5% on average, and an increase of the planetary boundary layer height by around 100 m. When compared to observations, the urbanized simulations performed better for all of the considered meteorological variables.

The modeled concentrations of both evaluated air pollutant species, PM$_{10}$ and PM$_{2.5}$, decreased by 35% and 28% on average, respectively, when using the urbanized configuration as compared to the BULK configuration. The main difference was caused by the reduction of primary aerosol. Compared to the measurements, the urbanized model results were underestimated by up to 40%. From the statistical evaluation of both model results, it could be concluded that the urbanized scheme led to an enhancement of the overall model error with an improvement only in the better representation of the diurnal variations. This important conclusion must be, however, considered with caution. It pointed out that there were multiple sources of model-observation disagreement, and besides the accurate representation of the urban canopy, e.g., high quality emission data were also crucial for the modeling of the urban air chemistry. Thus, model improvements aiming at better performance should consider every potential source of model errors. Otherwise, improving only one aspect (like including the urban effects) could lead to some worsening of the model accuracy.

It has to be noted that our conclusions were based on a winter episode, so they cannot be simply extended to other air pollution situations (e.g., occurring during summer); however, they confirmed previous findings about the urbanization effect on air quality during winter, and thus, they revealed some general behavior of winter air pollution in cities regardless of the region or the particular air pollution situation. It also has to be stressed that the "offline" nature of the coupling between WRF and CAMx brought some errors to the air quality predictions, as pointed out by Grell [68], Baklanov [69], especially regarding vertical mass distribution. Indeed, it was largely influenced by the choice of the urban canopy scheme in our study via turbulence. On the other hand, offline coupling here allowed independent development of the coupled components and the implementation of new parameterizations.

**Author Contributions:** Conceptualization, P.H. and K.E.; methodology, P.H., J.K., and J.G.; validation, J.Ď.; formal analysis, J.Ď.; investigation, P.H. and K.E.; data curation, N.B.; writing, original draft preparation, J.Ď., P.H., and K.E.; writing, review and editing, P.H., J.Ď., M.B., J.K., J.G., and O.V.; visualization, J.Ď. and O.V.; supervision, P.H., O.V., and T.H.; project administration, T.H. and O.V.; funding acquisition, T.H. All authors read and agreed to the published version of the manuscript.

**Funding:** This work was done within the project URBI PRAGENSI—"Urban Development of weather forecast, air quality, and climate scenario for Prague" financed by the Operational Programme Prague –Growth Pole of the Czech Republic, Project No. CZ.07.1.02/0.0/0.0/16_040/0000383, with the partial support of the SVV260581project supported by Charles University.

**Acknowledgments:** The authors would like to thank Petra Bauerová for providing the processed data from ceilometers.

**Conflicts of Interest:** The authors declare no conflict of interest.

**Appendix A. The Definition of the Statistical Scores**

Bias (mean bias):

$$bias = \frac{1}{N}\sum_{i=1}^{N} c_i - o_i$$

RMSE (Root Mean Squared Error):

$$RMSE = \sqrt{\frac{1}{N}\sum_{i=1}^{N}(c_i - o_i)^2}$$

r (correlation):

$$r = \frac{\sum_{i=1}^{N}(c_i - \bar{c})(o_i - \bar{o})}{\sqrt{\sum_{i=1}^{N}(c_i - \bar{c})^2}\sqrt{\sum_{i=1}^{N}(o_i - \bar{o})^2}}$$

$N$ denotes the number of samples, and $c$ and $o$ stand for modeled and observed values.

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
