# Peer review of "High Resolution Air Quality Forecasting over Prague within the URBI PRAGENSI Project: Model Performance during the Winter Period and the Effect of Urban Parameterization on PM"

_atmosphere, doi:10.3390/atmos11060625_

Round 1

Reviewer 1 Report

The manuscript aims at presenting the performance of the high resolution air quality forecasting system in the city of Prague testing the effect on PM10 and PM2.5 concentrations of two different urban parameterization in the model WRF, namely using the BEP+BEN urban canopy scheme in comparison with the BULK treatment.

This kind of analysis is undoubtedly worthy of attention, but, in my opinion, I would suggest accepting with major revisions the manuscript before the possible publication after some improvements and after the questions raised below had been addressed by the authors.

General comment

The work presents the results of the high resolution air quality forecasting model in the city of Prague testing two different WRF parameterization of the urban canopy scheme in reproducing a specific winter smog event (January and February 2017). The event has not been described even in the supplementary material just to give an idea of the event registered. In my opinion, the work is an exercise aimed at preliminary estimating the sensitivity of the chosen Chemical Transport Model (CAMx) at two different WRF parametrization. More robust and longer-term statistics are needed before determining which of the two different choices improve the air quality results.

Moreover, the results are often commented in a descriptive way without a real scientific discussion in analyzing the results. The authors should deepen the discussion about possible interpretation of the different obtained results. 

Please, could you provide information about the dataset used? Which kind of dataset they're considering (e.g., size, unstructured data, ... )? There is lack of dataset info, that can be maybe added as supplementary material.

No details about how the simulations have been run are provided (for example, parallelization of the algorithms/software).

Main comments

No enough technical details are provided for a possible reproducibility of the study; for example, in paragraph 2.1.1 (WRF), please give more information on the meteorological simulations. Add more info about initial and boundary conditions.

The authors should give also some more details about the typical PM levels during the episode chosen in the study.

It could be useful to introduce a table both for WRF (para 2.1.1) and CAMx (para 2.1.2) to describe the configuration used for the model's setup.

Information about the observations used needs a more in-depth discussion (for example, are they from the Airbase DB? Are all validated?...)

Line 179: how was the best candidate height selected?

Line 237: it could be useful to insert an Appendix with details on the formulation of skill scores

Line 242: why is RMSE reduced in the BEP+BEM for T2? From table 1 they are the same

Table2: please calculate the RMSE of the daily average of the whole period considered, more information could be provided and explained

Figg. 4, 6, 7, 8: it could be useful to have also PM observed and simulated (in both parametrization) daily concentrations to have an idea of model behaviour in the whole considered period

in the whole period to analyse the model behaviour in the entire selected period

Line 313: more explanations or interpretations should be provided at this point because differences are significant

Minor comments.

Line 18: please put some references for the sentence

Line 35: “by many that” -> “in many studies that”

Line 69-72: please rephrase. If you have already identified the main sources of pollution you are somehow talking about specific pollutants, so rephrase the sentence.

Line 81: how many days of forecasts are considered in the modeling system used? It has not been specified in the manuscript

Line 124: spell out all acronyms. In this case: ‘CMAQ’. But check in other parts of the manuscript (other examples, Line 141 – ATEM; Line 147 – CAMS, etc).

Line 145: ref for the LIFE project

Line 148: why has the version v1.1 of the CAMS anthropogenic emissions been used? There are more recent version

Line 149-150: SO2 and NH3 use subscript for “2” and “3” -> SO2, NH3

Line 183: please, insert more info about the vertical level heights of the model

Language

Please check the language, there are few mistakes to be corrected. Some examples are listed here:

Line 36: has → have

Line 93-65: the sentence is not clear, please rephrase

Line 122: eliminate coma

Author Response

The manuscript aims at presenting the performance of the high resolution air quality forecasting system in the city of Prague testing the effect on PM10 and PM2.5 concentrations of two different urban parameterization in the model WRF, namely using the BEP+BEN urban canopy scheme in comparison with the BULK treatment.

This kind of analysis is undoubtedly worthy of attention, but, in my opinion, I would suggest accepting with major revisions the manuscript before the possible publication after some improvements and after the questions raised below had been addressed by the authors.

Authors' response: We appreciate the time and effort taken to provide feedback on this manuscript. We are grateful for the detailed comments received which have guided us to make significant improvements to the paper. We will do our best to consider the reviewer’s each concern and our point-by-point responses follow including the action taken within the manuscript.

General comment

The work presents the results of the high resolution air quality forecasting model in the city of Prague testing two different WRF parameterization of the urban canopy scheme in reproducing a specific winter smog event (January and February 2017). The event has not been described even in the supplementary material just to give an idea of the event registered. In my opinion, the work is an exercise aimed at preliminary estimating the sensitivity of the chosen Chemical Transport Model (CAMx) at two different WRF parametrization. More robust and longer-term statistics are needed before determining which of the two different choices improve the air quality results.

Authors’ response: We agree that the chosen air pollution event should have deserved a more detailed description. We extended the paragraph describing meteorological and air quality situation to the subsection: Simulated period (2.2.2). Regarding the robustness of the results, the aim of the study was not to make general conclusions about which model setup performs better for any air pollution situation, but rather to show how sensitive the model predictions are during situations when accurate model predictions are essential. Indeed, a PM pollution episode as presented in the study, belongs to such situations and our results showed that the model predictions are marked with relatively large uncertainty lying in the chosen parameterization of the urban canopy layer.

The results offer preliminary insight into the accuracy of the meteorology/air-quality predictions for Prague (as indicated by the title), and as such, we found it crucial to demonstrate the model strengths and weaknesses for extreme air pollution events. We plan a follow-up study, where the focus will be on long term model performance with (hopefully) conclusive results about the general behaviour of different model configurations with regards to the accuracy of the predicted driving meteorological conditions and air pollutant concentrations. This is clarified in more detail in the revised manuscript.

Moreover, the results are often commented in a descriptive way without a real scientific discussion in analyzing the results. The authors should deepen the discussion about possible interpretation of the different obtained results. 

Authors’ response: We agree with the reviewer that the results could be discussed in more detail. There are model deviations from measured values as well as differences between individual model predictions that are not sufficiently commented/interpreted. We tried to include more explanation in the discussion section including a larger number of already published material that helps in the interpretations of the documented results. We stress here that in the Results section, we try to offer only the pure presentation/description of the obtained results and it is the Discussion section where we offer in depth scientific explanation for the modeled biases and inter-model differences.

Please, could you provide information about the dataset used? Which kind of dataset they're considering (e.g., size, unstructured data, ... )? There is lack of dataset info, that can be maybe added as supplementary material.

Authors’ response: All the input data used in this study are mentioned and cited in the manuscript. These are: landuse information including urban parameters compiled for the city of Prague which now encompasses a whole paragraph (lines 222-238) with all the details regarding the source data. Other data used are the meteorological and chemical boundary conditions, the raw emissions inputs, the total ozone column data and the observation data from Prague used for the comparison. 

No details about how the simulations have been run are provided (for example, parallelization of the algorithms/software).

Author's response: The simulations are performed on an Intel Xeon-based HPC (High Performance Computing) cluster at the Faculty of Mathematics and Physics, with parallelization using the OpenMPI over OmniPath technology and GNU compilers. Added at the end of sec. 2.2.1 (lines 250-253).

Main comments

No enough technical details are provided for a possible reproducibility of the study; for example, in paragraph 2.1.1 (WRF), please give more information on the meteorological simulations. Add more info about initial and boundary conditions.

Authors’ response:  Initial and boundary conditions for the WRF predictions are taken from the GFS model results provided by NCEP at 0, 6, 12 and 18h UTC, further details added in the section 2.2.1.

 The authors should give also some more details about the typical PM levels during the episode chosen in the study.

Authors’ response: observed PM concentrations were added to the section describing the meteorology situation of the selected period (section 2.2.2, lines 255-280).

It could be useful to introduce a table both for WRF (para 2.1.1) and CAMx (para 2.1.2) to describe the configuration used for the model's setup.

Authors’ response: We added a table (Table 1.) listing the model and domain configuration for both WRF and CAMx. Related components of the both models are listed next to each other (like WRF microphysics vs. CAMx wet deposition).

Information about the observations used needs a more in-depth discussion (for example, are they from the Airbase DB? Are all validated?...)

Authors’ response: The measurements were acquired from the database of the Czech Hydrometeorological Institute. These data are thoroughly validated and so only valid observations were used. We added this information in the text (lines 169-175).

Line 179: how was the best candidate height selected?

Authors’ response: The best candidate height is determined automatically (using quality indices) by the software for each of the individual measurements. For each hourly mean, the heights with the highest sum of quality indices were averaged. We have updated the text to better explain this process (lines 177-184).

Line 237: it could be useful to insert an Appendix with details on the formulation of skill scores:

Authors’ response: We added and Appendix (A) with the definition of the individual skill scores used in the study

Line 242: why is RMSE reduced in the BEP+BEM for T2? From table 1 they are the same

Authors’ response: Indeed, the RMSE is reduced in the BEP+BEM simulation, however the reduction occurred over the 2nd decimal points and the number in the tables were rounded to 1 decimal point only. We reformulated the sentence to reflect this, i.e. that RMSEs are about the same.

Table2: please calculate the RMSE of the daily average of the whole period considered, more information could be provided and explained

Authors’ response: We have included the statistics also for the daily averages of PM concentrations in Table 3. The results for the daily averages are very similar to the hourly means. We have added some information about the day-to-day variations.

Figg. 4, 6, 7, 8: it could be useful to have also PM observed and simulated (in both parametrization) daily concentrations to have an idea of model behaviour in the whole considered period in the whole period to analyse the model behaviour in the entire selected period

Authors’ response: We have added the plots of daily concentrations (both PM fractions and their components: Fig. 7, 10, 12, 14) as well as the statistical evaluation of the daily averages.

Line 313: more explanations or interpretations should be provided at this point because differences are significant

Authors’ response: Indeed, the differences are significant and we well understand the physical explanation behind them: it is the enhanced vertical turbulent diffusion in the BEP+BEM scheme compared to the BULK approach (in general, turbulence over urban areas is strong and its magnitude cannot be resolved in BULK schemes). We discuss these differences in greater detail in the Discussion.

Minor comments.

Line 18: please put some references for the sentence

Authors’ response: reference added.

Line 35: “by many that” -> “in many studies that”

Authors’ response: Corrected.

Line 69-72: please rephrase. If you have already identified the main sources of pollution you are somehow talking about specific pollutants, so rephrase the sentence.

Authors’ response:  Rephrased (line 69).

Line 81: how many days of forecasts are considered in the modeling system used? It has not been specified in the manuscript

Authors’ response: Yes, this information is not provided in the model description section, we find it more appropriate in the Experiment setup section, where the timeline of the simulations is described (lines 195, 240).

Line 124: spell out all acronyms. In this case: ‘CMAQ’. But check in other parts of the manuscript (other examples, Line 141 – ATEM; Line 147 – CAMS, etc).

Authors’ response: We have checked all acronyms and added the information. ATEM comes from the Czech name of the company. Besides its translation, we have also included the full name in Czech to indicate the origin of the acronym.

Line 145: ref for the LIFE project

Authors’ response: The project is referenced in the text by the project number. We also added a link to the project website.

Line 148: why has the version v1.1 of the CAMS anthropogenic emissions been used? There are more recent version

Authors’ response: This version was the latest available at the time when the forecast system was set up. The same configuration was therefore used in this case study. Within the future development of the modeling system, the input emission data are planned to be updated.

Line 149-150: SO2 and NH3 use subscript for “2” and “3” -> SO2, NH3

Authors’ response: Corrected.

Line 183: please, insert more info about the vertical level heights of the model

Authors’ response: This information is added for both WRF and CAMx. (lines 189-190)

Language

Please check the language, there are few mistakes to be corrected. Some examples are listed here:

Line 36: has → have

Authors’ response: the verb “has” here belongs to the “model treatment” i.e we kept it in singular form and rephrased the sentence for better clarity.

Line 93-65: the sentence is not clear, please rephrase

Authors’ response: Rephrased.

Line 122: eliminate coma

Authors’ response: Corrected.

Reviewer 2 Report

Dear Authors,

My review on your manuscript is attached as a pdf file.

Reviewer

Author Response

The authors evaluated a WRF/CAMx forecast modeling system set up for Prague, Czech Republic. They performed a simulation for a winter pollution episode and compared to a non-urbanized run with BULK treatment. This is an interesting paper.

Authors' response: We would like to thank the reviewer for their thorough consideration of our manuscript and for all concerns raised and suggestions for improvements. We will consider each of them and our point-by-point responses follow:

My comments are listed below

Line 71: correctly: NO2 instead of NO2.

Authors’ response: Corrected

Section 2.1.3. Emissions: 1st paragraph: What is the error of the model due to the

fact that the emission data are from three different years (2015, 2016, 2017)? Did you

perform any error analysis?

Author's response: We used the latest and most detailed emission data available at the time when the forecast system was set up for the chosen air pollution episode with the expectation that the year to year variation of anthropogenic emissions is small or at least well within the overall model uncertainty. Based on the emission reporting data (http://portal.chmi.cz/files/portal/docs/uoco/oez/EBCRREZZO1-4Kraj2008-2018.xls) for Prague (CZ010), the variation of emission totals between the three years reaches a maximum of 10% for most components. 

Section 2.1.3. Emissions: 1st paragraph: What is the uncertainty of the model due to the fact that emission data with different resolutions and different proportions of grid point / point source / line source data were included in the model from area to area?

Authors’ response: In section 2.1.3 a detailed description of emissions used is provided. As mentioned in the comment, these are formed by different source types (point, line, area), which are then gridded into the CAMx grid by the FUME model (source geometry is intersected with CAMx grid and spatial-weighted average is calculated). The data used for the Czech Republic have the highest spatial resolution available and therefore provide highest possible accuracy. 

2.1.4. Air quality stations: meteorology and pollutants, 1st paragraph: What is the error

/ uncertainty of the model evaluation, including that e.g. you compare point source

data with grid point data.

Authors’ response: We aimed for maximal comparability of the measured and modelled data. For pollutant concentrations this meant taking the model concentration from the gridbox that lies above the measurement station, which is the most correct approach as CAMx considers uniform concentrations for the whole gridbox. On the other hand, we took for WRF the closest gridpoint to the measured site. Spatial interpolation would be maybe more correct, but we chose this approach to ensure that for the particular station we chose a model gridpoint that has the same urban parameters, i.e. the closest one. The uncertainty is related to the spatial representativeness of the measurement station, which in this case - due to the use of background stations only - should be comparable to the model resolution.

Section 2.2. Experimental setup, first sentence: WRF (version 4.0.3) was run on three nested domains with horizontal resolutions of 9 km, 3 km and 1 km, with 173 x 153, 169 x 151 and 84 x 84 grid points, respectively (Fig. 2). How to ensure comparability between these three height levels with different resolutions? If we do compare, what will be its error?

Authors’ response: Indeed, the finer the resolution the more detail of the terrain with more elevated features are resolved by the model. This means that for a particular gridbox, the model terrain elevation is, in general, different. This is a common “feature” of nested runs and should be accounted for when comparing results for surface variables (e.g. near surface temperature) from different nesting levels corresponding to different resolutions. This paper aimed at Prague only so all the results are taken from the inner domain and no comparison with the coarser model data is made. However, even if we did such a comparison, given that the terrain in and around Prague is not so variable, we believe that the model results would be very similar with exactly the same conclusions made as if data were taken only from the 1 km x 1 km fine scale domain.

Table 1: The Pearson linear correlations themselves are not very interesting for us.

Instead, I suggest you write p-values (which is the probability of rejecting the true 0-

hypothesis).

Authors’ response: We have added p-values to Table 2 and 3 to indicate if the correlation is statistically significant. In all cases for both meteorological variables and PM concentrations, the p-value was much smaller than 0.01, confirming statistically significant correlation between the measurements and simulations.

line 273: correctly: Table 2 , instead of Tab. 2 , throughout the whole manuscript.

Table is never abbreviated.

Authors’ response: Corrected in all occurrences.

Table 2: Regarding the Pearson linear correlations, see Table 1.

Authors’ response: See above.

line 308: correctly: (PNO3), (PSO4) and (PNH4) throughout the whole manuscript,

instead of (PNO3), (PSO4) and (PNH4);

Authors’ response: Corrected in all occurrences.

Section 5. Conclusions , paragraph 2: When running the urbanized vs non-urbanized

simulations, the received average difference in temperature at 2 m, in wind speed, and

in relative humidity and boundary layer height proved to be 2 C, 0.5 m s-1, 5% and

100 m, respectively. Have any calculations been made that these are significant

Differences?

Authors’ response: Paired t-test was performed for the corresponding time series for each meteorological variable studied and it showed statistically significant differences on at least 99% level of significance. Moreover, the differences are in line with the expected behaviour of a more comprehensive urban scheme with respect to a simple bulk approach (see Discussion for previous studies confirming the same conclusions).

Section 5. Conclusions , paragraph 3: You write here that the modeled average

concentrations of PM10 and PM2.5 decreased by 35% and 28%, respectively when

using the urbanized vs BULK simulations. The question is the same, as above: Have

you made any calculations, whether these numbers are significant differences?

Authors’ response: We performed a significance test for PM as well (using a paired t-test) which showed that the differences are statistically significant on at least 99% level. Moreover, the temporal evolution of daily averages showed that the differences between the two configurations are systematic and detectable during each simulated day.

Reviewer 3 Report

Based on the WRF/CAMx modeling framework, this study investigates the impacts of urban land parameterization (BULK v.s. BEP+BEM) on predicted meteorology and PM concentrations over Prague. Overall, this manuscript is well written, and the topic of this manuscript is popular and important. However, the authors should carefully address my comments below before I’d like to recommend it to be published.

  1. In Line 187, why BULK scenario is described as “non-urbanized”, confusing. The SLAB urban scheme in WRF does treat the urban land differently, but in a highly simplified way. I mean, whether “non-urbanized” may depend on the exact land use or land cover (LULC) data you used. However, no detailed description about the used LULC data is found. LULC is the key parameter in this study, and the authors should draw a map plot illustrating the LULC conditions in the two scenarios.
  2. In Line 190, the configuration of BEP+BEM, especially the detailed key parameters used, is far from well described. BEP+BEM is much more complicated than the SLAB or single-layer urban canopy model (SLUCM), and needs much more input parameters (especially parameters relevant with urban morphology). In Figure 5, as explained by the authors, in BEP+BEM scenario the vertical mixing is much stronger, leading to an obviously underestimated PM concentrations at surface layer. I doubt the correctness or representativeness for the input parameters in BEP+BEM scenario. I mean, the authors should make more efforts to validate that the BEP+BEM is properly used with realistic input parameters.
  3. In Figure 2, it might be better if the outset domain should occupy much more area fraction in the plot.
  4. In Figure 4, the observed diurnal cycle for PM is normal, but the simulated diurnal cycle for PM is quite surprising and confusing for me especially in BULK scenario. Please refer to Figure 3, the simulated diurnal cycles for PBLH and RH are both quite normal, and well match the observed diurnal cycle for PM. I doubt if there is any problem for the coupler between WRF and CAMx? Or due to the diurnal cycle in simulated emissions? Neither BULK nor BEP+BEM scenario could provide a relatively satisfying simulation for PM.
  5. As using an offline model, the authors should mention in the end of the manuscript that one of the limitations for this study is that, the feedbacks between urban meteorology and urban air pollution are ignored. If using an online model, e.g., WRF/Chem, such issue could be resolved.

Author Response

Based on the WRF/CAMx modeling framework, this study investigates the impacts of urban land parameterization (BULK v.s. BEP+BEM) on predicted meteorology and PM concentrations over Prague. Overall, this manuscript is well written, and the topic of this manuscript is popular and important. However, the authors should carefully address my comments below before I’d like to recommend it to be published.

Authors’ response: We would like to thank the reviewer for their thorough consideration of our manuscript and for all suggestions for improvement and correction. We will consider each of them and our point-by-point responses follow:

In Line 187, why BULK scenario is described as “non-urbanized”, confusing. The SLAB urban scheme in WRF does treat the urban land differently, but in a highly simplified way. I mean, whether “non-urbanized” may depend on the exact land use or land cover (LULC) data you used. However, no detailed description about the used LULC data is found. LULC is the key parameter in this study, and the authors should draw a map plot illustrating the LULC conditions in the two scenarios.

Authors’ response: We agree here that our “definition” of non-urbanized version/configuration is not completely clear as even in the BULK (SLAB) scheme urban landuse was indeed considered. We will clarify this in the description of the experiment: i.e. we prefer to use non-urbanized version in the sense that the bulk scheme does not account for the 3D nature of the urban weather phenomenon and the surface is considered in the same “flat” manner than any other non-urban surfaces. Hence, it is a “non-urbanized” approach. We also added more detailed information on the urban landuse data used to configure the BEP+BEM scheme.

In Line 190, the configuration of BEP+BEM, especially the detailed key parameters used, is far from well described. BEP+BEM is much more complicated than the SLAB or single-layer urban canopy model (SLUCM), and needs much more input parameters (especially parameters relevant with urban morphology). In Figure 5, as explained by the authors, in BEP+BEM scenario the vertical mixing is much stronger, leading to an obviously underestimated PM concentrations at surface layer. I doubt the correctness or representativeness for the input parameters in BEP+BEM scenario. I mean, the authors should make more efforts to validate that the BEP+BEM is properly used with realistic input parameters.

Authors’ response: 

In the revised manuscript, we provided a very detailed list of the input data used to create the most realistic urban land-cover input for BEP+BEM scenario. First of all, for both configurations, land-cover information is taken from the high resolution (100 m) CORINE CLC 2012 data (https://land.copernicus.eu/pan-european/corine-land-cover).

For Prague, the land-cover info including urban parameters, like street geometries, albedoes, emissivities, vegetation fraction etc. are adopted from multiple high resolution data sources. Three landcover sources were combined into one spatially resolved dataset: the Urban Atlas 2018 (Copernicus Land Monitoring Services, Urban Atlas (https://land.copernicus.eu/local/urban-atlas/urban-atlas-2018?tab=metadata), ZABAGED geodatabase (The Fundamental Base of Geographic Data of the Czech Republic; The Czech Office for Surveying, Mapping and Cadastre: https://geoportal.cuzk.cz/) and digital technical map of Prague provided by GeoPortal Prague (https://www.geoportalpraha.cz/en). Further, landcover fractions without natural vegetation were prepared using the European Settlement Map (release 2017; Copernicus Land Monitoring Services,  https://land.copernicus.eu/pan-european/GHSL/european-settlement-map/esm-2012-release-2017-urban-green?tab=metadata), for detailed building information, the 3D model of Prague was used (3D model; GeoPortal Prague; https://www.geoportalpraha.cz/en), information about streets geometries are based on ZABAGED. Vector data provided by the listed sources were aggregated to gridded data with 333 m spatial resolution. Emissivities of different urban surfaces are based on ASTER Global Emissivity Dataset with ~100m spatial resolution (ASTERT GED; https://emissivity.jpl.nasa.gov/aster-ged) and the Land Surface Emissivity algorithm for LANDSAT-8 Gillespie et al.(2014). In summary, urban parameters are considered as 2D arrays instead of constant values, i.e. every urban grid-point has an unique combination of urban parameters allowing to capture the spatial distributions urban effects more precisely. 

Regarding the PM underestimation, it is stronger in the BEP+BEM scenario, but the main reason is probably the underestimated emissions. Indeed, our comparison with measured PBL heights showed that the BEP+BEM reproduced PBL better than the BULK approach. Consequently, the PM underestimation is attributable to underpredicted input of material into the lowermost model layers (i.e. emissions).

In Figure 2, it might be better if the outset domain should occupy much more area fraction in the plot.

Authors’ response: we zoomed in the picture. The model master domain now occupies most of the picture.

In Figure 4, the observed diurnal cycle for PM is normal, but the simulated diurnal cycle for PM is quite surprising and confusing for me especially in BULK scenario. Please refer to Figure 3, the simulated diurnal cycles for PBLH and RH are both quite normal, and well match the observed diurnal cycle for PM. I doubt if there is any problem for the coupler between WRF and CAMx? Or due to the diurnal cycle in simulated emissions? Neither BULK nor BEP+BEM scenario could provide a relatively satisfying simulation for PM.

Authors response: Indeed, the PBL height and wind speed are modelled with reasonable accuracy so we concluded, that the reason for the incorrect model representation of PM diurnal cycle (especially of PM10) lies in the diurnal time-disaggregation factors used in our study (Gon et al.(2011)). Probably the particle resuspension from vehicle emissions is overestimated leading to the prolongation of the emissions peaks during morning and evening rush hours in CAMx. Revision of diurnal emission factors (e.g. by using traffic density temporal profile measurements from Prague) should be performed in the future. This is discussed in the revised manuscript too.

As using an offline model, the authors should mention in the end of the manuscript that one of the limitations for this study is that, the feedbacks between urban meteorology and urban air pollution are ignored. If using an online model, e.g., WRF/Chem, such issue could be resolved.

Authors’ response: we agree that an offline couple brings some inconsistencies between the full meteorological fields provided by WRF at every timestep and the actual driving meteorology that feeds CAMx (via the WRFCAMx preprocessor). We added a few thoughts on this issue with relevant references at the end of the Conclusions section.

Round 2

Reviewer 1 Report

The authors did a great effort in introducing all the points raised and in answering all the comments. There are some minor spelling errors to check (i.e RMSE in the Appendix A, substitute “sqaure” with “square”), but, in my opinion, the paper could be now accepted for publication in its revised version.

Reviewer 3 Report

The authors have carefully resolved my comments and improved the manuscript. I don't have new comments.